# ColVO: Colonoscopic Visual Odometry Considering Geometric and Photometric Consistency

## ABSTRACT

Locating lesions is the primary goal of colonoscopy examinations. 3D perception techniques can enhance the accuracy of lesion localization by restoring 3D spatial information of the colon. However, existing methods focus on the local depth estimation of a single frame and neglect the precise global positioning of the colonoscope, thus failing to provide the accurate 3D location of lesions. The root causes of this shortfall are twofold: Firstly, existing methods treat colon depth and colonoscope pose estimation as independent tasks or design them as parallel sub-task branches. Secondly, the light source in the colon environment moves with the colonoscope, leading to brightness fluctuations among continuous frame images. To address these two issues, we propose ColVO, a novel deep learning-based Visual Odometry framework, which can continuously estimate colon depth and colonoscopic pose using two key components: a deep couple strategy for depth and pose estimation (DCDP) and a light consistent calibration mechanism (LCC). DCDP utilization of multimodal fusion and loss function constraints to couple depth and pose estimation modes ensures seamless alignment of geometric projections between consecutive frames. Meanwhile, LCC accounts for brightness variations by recalibrating the luminosity values of adjacent frames, enhancing ColVO's robustness. A comprehensive evaluation of ColVO on colon odometry benchmarks reveals its superiority over state-of-the-art methods in depth and pose estimation. We also demonstrate two valuable applications: immediate polyp localization and complete 3D reconstruction of the intestine. The code for ColVO is available at https://github.com/xxx/xxx.

## CCS CONCEPTS

• **Computing methodologies → Computer vision tasks**.

## KEYWORDS

Visual Odometry, Monocular Depth Estimation, Pose Estimation, Colonoscopy

## 1 INTRODUCTION

The primary objective of colonoscopy examinations is lesion localization. Traditional colonoscopy procedures typically rely on capturing two-dimensional (2D) images from colonoscopy videos to identify the location of lesions within the planar region. However, due to limitations in 2D observation such as restricted field-of-view

*Conference ACM MM'24, Oct.28-Nov.1, 2024, Melbourne, AUS*

© 2018 Copyright held by the owner/author(s). Publication rights licensed to ACM.
ACM ISBN 978-1-4503-XXXX-X/18/06
https://doi.org/XXXXXXX.XXXXXXX

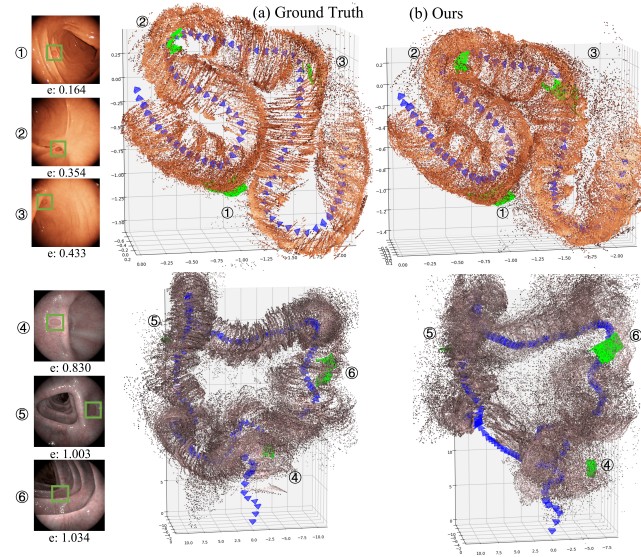

**Figure 1: Visualization comparison of 3D reconstruction and polyp localization between GT (left) and predicted results (right). The results from ColVO are close to GT. *e* denotes the error between the estimated 3D position and the GT position of the polyp.**

(Fov) and potential occlusions, physicians often rely on their expertise to estimate the approximate location of lesions within the intestinal tract, making it difficult to avoid subjective errors. With the emergence of large-scale cross-modal colonoscopy datasets and integrated deep learning (DL) methods, DL-based three-dimensional (3D) perception techniques have the potential to enhance lesion localization accuracy by recovering 3D spatial information of the intestine. However, most existing methods [19, 21, 24, 40] focus only on single-image depth estimation, overlooking the geometric continuity between frames of colonoscopy videos. Consequently, they can only reflect the spatial position of lesions within a local frame and cannot accurately determine their precise global coordinates within the entire colon. To solve this problem, some work [12, 29, 30, 32, 38] have attempted to joint camera motion estimation with depth estimation by minimizing photometric errors between consecutive frames to obtain global 3D spatial information of the intestine. However, these methods fail to consider *specific challenges* in the colon, including sparse texture features and lighting fluctuations. The former affects the sufficient and effective feature extraction in network models, thereby impacting depth and pose estimation consistency and accuracy. The latter arises from the correlated motion between the camera and light source during colonoscopy examinations and the non-Lambertian reflectance properties of tissues, both of which significantly affect the constraints of photometric errors on network

models. Consequently, these methods cannot balance the accuracy of depth and pose estimation.

In this paper, we propose ColVO, a novel colonoscopic visual odometry framework for precise spatial localization of lesions by simultaneously enhancing depth and camera motion estimation in colon environments. To address the challenges in the colon, we first introduce the deep couple strategy for depth and pose estimation (DCDP). DCDP incorporates cross-modal RGB and inferred depth features from the depth estimator through multilevel fusion in the pose estimator. The estimated depth and pose are jointed through a reprojection-based photometric loss function to constrain the network. By leveraging the scene geometry features from the depth maps, DCDP improves pose estimation accuracy and ensures geometric consistency in low-texture conditions. To tackle the illumination variation in the intestinal environment, we present the light consistent calibration mechanism (LCC). The LCC calculates the light state of each frame based on the surface normal and camera pose. By reconstructing the photometric loss function using LCC and mask for non-Lambertian regions, photometric consistency between adjacent frames can be aligned to enhance the continuity and accuracy of depth and pose estimation under nonuniform illumination conditions. The main contributions of our paper are as follows:

- We design a novel VO framework tailored for lesion localization in colonoscopy (ColVO) that estimates depth and camera motion simultaneously. To meet the clinical application of precise polyp localization and 3D colon reconstruction, our ColVO framework focuses on geometric consistency and photometric consistency to enhance the accuracy of depth and pose estimation.
- We introduce DCDP, a novel technique that couples depth and pose estimation in a single network and uses cross-modal fusion of RGB and depth features to enhance the pose estimator.
- We propose LCC, a novel technique that calibrates the pixel values of adjacent frames based on the light state and masks out the non-Lambertian regions to enhance the photometric consistency.
- We conduct experiments on two synthetic datasets and one real dataset and show that ColVO achieves state-of-the-art accuracy in depth and pose estimation in the colon environment. We also show that ColVO enables complete and coherent 3D reconstruction of the colon, accurate colonoscope trajectory prediction, and precise lesion localization.

## 2 RELATED WORK

### 2.1 Monocular Depth Estimation Method

Traditional depth sensors such as stereo, structured light, or time-of-flight cameras are ineffective when used in narrow intra-body environments, especially in colon scenes [30]. Therefore, algorithm-based monocular depth estimation using colonoscopic images is preferred. However, these methods encounter challenges in colon environments, including lighting variations and texture scarcity. Oda *et al*. [21] addressed the issue of light reflections and textures in monocular endoscopic depth estimation by directly removing them using Lambertian surface translation. Similarly, Visentini-Scarzanella *et al*. [33] employed a mapping technique for input bronchoscopy frames. However, these crude approaches hinder the network model from effectively capturing the textured features of the colon and fail to address the fundamental challenges posed by the intra-body

environment in depth estimation tasks. The scarcity of texture-rich images makes it challenging to provide valuable information for guiding network training. Although some studies [3, 8, 10, 13–16] have considered using sparse depth information, easily obtained through traditional structure-from-motion or simultaneous localization and mapping (SLAM) methods, to assist in image-based depth estimation. However, these methods introduce challenges such as inconsistent distribution of valid pixels when combined with full-resolution RGB images. Inspired by depth modality's effectiveness in enhancing depth estimation, our method leverages the fusion of inferred depth with RGB information to improve depth and pose estimation in the colon.

Although existing DL-based depth estimation algorithms can generate satisfactory depth maps for individual images, their effectiveness in endoscopy is often limited to the local spatial region of a single frame [19, 24, 42], disregarding the correlation between consecutive frames. Consequently, when attempting to stitch together the depth maps of each frame to generate a complete 3D model of the colon, misalignment may occur in the space projection of adjacent frames, resulting in deficiencies in the reconstructed gastrointestinal model and the inability to obtain the location of lesions within the entire colon.

### 2.2 Joint Depth and Pose Estimation Methods

Localization lesion in the colon requires predicting both dense depth and camera pose for each frame. To achieve this, some works design VO framework jointly training depth estimation network (DepthNet) and pose estimation network (PoseNet). Turan *et al*. [30] proposed Deep EndoVO, which is the first monocular VO approach for endoscopy through DL techniques. However, Deep EndoVO does not strictly belong to the end-to-end VO as the depth map is derived from traditional visual geometry methods. Similarly, DL technology is only adopted in the DepthNet of Endo-Depth-and-Motion [25], resulting in a disjoint between DepthNet and the Pose estimator and preventing close integration during training. Subsequently, some studies have utilized view synthesis between consecutive frames as a photometric error function to constrain both depth and pose, thereby achieving unsupervised/self-supervised VO. The premise for photometric error is constant illumination and unchanged pixel brightness with respect to the view direction. Unsupervised VO must incorporate illumination variations in the endoscopy environment. Turan *et al*. [32] presented the first unsupervised endoscopic VO Endo Odometry Learner and designed a soft reliability mask to address factors that disrupt view synthesis, such as non-Lambertian surfaces. However, [32] lacked sufficient details on the soft reliability mask. Ling *et al*. [12] also created masks to remove highlighted regions according to the values of saturation and intensity. Zhang *et al*. [45] extended classic photometric loss with feature matching to compensate for changing contrast and brightness. Ozyoruk *et al*. [22] leveraged brightness-aware photometric loss to improve the robustness under fast illumination changes in endoscopic videos. These methods primarily rely on masks or adjustments in lighting from the image surface to mitigate illumination effects. In contrast, our approach tackles the underlying cause of illumination variations in the intestinal environment, namely, the motion of the light source accompanying the camera, and recalibrates the illumination. Notably,

large illumination changes and a lack of texture in colon scenes can result in unsatisfactory performance of the DL when trained without supervision on colonoscopic videos [26, 38, 45, 46].

Current VO frameworks typically treat DepthNet and PoseNet as separate parallel networks and overlook the interconnectedness between the two sub-networks. In contrast, our ColVO leverages the potential benefits of the inferred depth map from DepthNet to improve pose estimation accuracy. Additionally, some endoscopic VO methods [12, 25, 30, 32] are not specifically designed for the colon. Some methods [18, 45] can only achieve local mapping to reconstruct the partial colon, thus failing to meet the demand for global lesion localization.

## 3 METHOD

In this section, we present our ColVO network which takes consecutive colon video frames as input and simultaneously generates dense depth maps and colonoscopic trajectories. We first describe the entire ColVO framework, and then explain the learning objectives of ColVO.

### 3.1 Method Overview and Network Architecture

Our goal is to jointly train DepthNet and PoseNet using monocular colonoscopic videos. As shown in Fig. 2, given two consecutive colon frames with dimensions $H \times W \times C$ at time $t$ and $t+1$, DepthNet first generates dense depth maps with dimensions $H \times W \times 1$. Then, the predicted depth maps, denoted as $(\hat{D}_t, \hat{D}_{t+1})$, are concatenated with the image pairs and fed into the PoseNet to estimate the relative pose parameterized as 6-DoF transformation matrices $\hat{T}_{t \to t+1}$ between consecutive frames. DepthNet follows the U-Net encoder-decoder architecture [27], while PoseNet adopts an encoder-regressor structure. The DCDP strategy leverages the RGB-depth incremental fusion in PoseNet for improved geometric consistency between consecutive frames. To deal with drastic illumination changes and weak texture in the colon, ColVO integrates both supervised and self-supervised constraints to train DepthNet and PoseNet. Supervised signals are derived from ground-truth (GT) depth maps and pose information, while self-supervised signals are based on photometric loss. As the movement of light sources with the colonoscope and non-Lambertian reflection of colon tissues readily disrupt the light consistency in the photometric loss, the LCC mechanism is designed for reconstructing the photometric loss to ensure consistent lighting conditions.

### 3.2 Deep Coupling Strategy for Depth and Pose Estimation

To achieve accurate lesion spatial localization, we need to estimate both dense depth and precise colonoscope trajectories from monocular images. However, these two tasks are interdependent and challenging in the colon environment. On one hand, the pose estimation relies on the image features, which are often limited and affected by the dynamic lighting in the colon. On the other hand, the depth estimation depends on the geometric consistency and continuity, which can be disrupted by the inaccurate pose estimation. Therefore, we propose a deep coupled VO framework that jointly trains DepthNet and PoseNet in a sequential and progressive manner.

Unlike the traditional parallel dual-stream structures [2, 6, 12, 30, 45, 47] that use only image inputs for pose estimation, our pose estimation sub-network leverages both RGB and inferred depth inputs from DepthNet. Inspired by Jiang $et$ $al$'s work [11], we use a multilayer incremental fusion strategy [11] to fuse cross-modal features from intermediate layers of the encoder. This strategy allows the PoseNet to benefit from the rich and relatively robust geometric features from DepthNet branch. We use ResNet-18 [43] as the encoder for both streams, but we only share the weights except for the batch normalization (BN) layer, thereby ensuring that the features before BN are in the same latent space, so that they can be exchanged between streams. To preserve complementary features between RGB and depth modalities, we use a channel exchange (CE) strategy [36] that swaps important feature elements in each stage of feature fusion.

$$E_1, E_2 = CE(BN(Relu(Conv(I, \hat{D})))) \qquad (1)$$

where $\hat{D} = DepthNet(I)$, $Relu$ is the activation function. After encoding each modality into the generic feature space, we use a pose regressor to predict a 6-dimensional relative pose representation from each feature.

$$\hat{T}_{s \to t} = PoseNet(E'_1, E'_2) \qquad (2)$$

where $E'_1$, $E'_2$ are generated by multiple Eq. 1 processes. The final output is the refined pose estimation from the fourth stage. We also apply the training loss proposed in [11] to prevent the model from reaching singular solutions.

### 3.3 Light Consistent Calibration

While the deep coupling of depth and pose estimators in the network architecture enhances the geometric consistency between consecutive frames, the DL-based VO task ensures photometric consistency between consecutive frames by minimizing photometric loss based on estimated depth $\hat{D}_t$ and pose $\hat{T}_{s \to t}$. In addition to the classical image pixel error [47], we also adopt image dissimilarity error SSIM [2, 37] in photometric loss to effectively handle illumination changes.

$$\mathcal{L}_p(I_t, \hat{I}_{s \to t}) = \alpha \frac{1 - SSIM(I_t, \hat{I}_{s \to t})}{2} + (1 - \alpha)\|I_t - \hat{I}_{s \to t}\| \qquad (3)$$

where $SSIM(I_t, \hat{I}_{s \to t})$ and $\|I_t - \hat{I}_{s \to t}\|$ stand for the element-wise similarity and pixel-level similarity between the target image $I_t$ and the synthetic target image $\hat{I}_{s \to t}$, respectively. $\hat{I}_{s \to t} = I_s \langle proj(\hat{D}_s, \hat{T}_{s \to t}, K) \rangle$, $\langle \cdot \rangle$ represents the sampling operator, $proj(\cdot)$ returns the 2D coordinates of the depths in $D_t$ when reprojected into the viewpoint of $I_s$. Following [2], we use $\alpha = 0.15$ in our framework.

As the classical VO systems task assumes a fixed absolute position of the light source in global environments, the synthesized images maintain constant illumination and unchanged pixel brightness with the target images. However, the light source moves with the camera in the colonoscopic videos, leading to the photometric variation of the same spatial position between consecutive frames (the upper right corner of Fig. 2). This means that a certain difference remains between $\hat{I}_{s \to t}$ and $I_t$ even if the depth and pose estimation are completely correct, which disfavors neural network learning. To address the issue of photometric inconsistency in ColVO, we propose an LCC mechanism considering light source mobility. LCC is defined as the ratio of luminosity generated by the reflection of light rays from the same spatial point on the colon surface between continuous time $t$ and $s$.

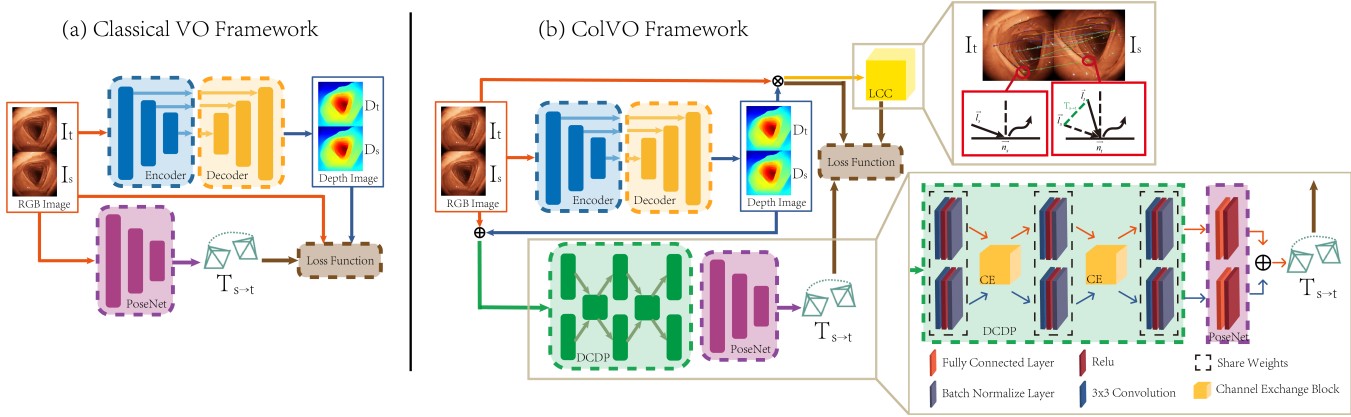

**Figure 2: Left: Classical DL-based VO framework with relatively separates DepthNet and PoseNet. Right: Overview of the proposed ColVO framework jointly with DepthNet and PoseNet in a sequential and progressive manner. DCDP also integrates inferred depth features with RGB to enhance the pose estimator. LCC recalibrates the luminosity values of adjacent frames caused by the light source mobility.**

$$LCC = \frac{m_t}{m_s} = \frac{e\vec{l_t}\rho\vec{n_t}}{e\vec{l_s}\rho\vec{n_s}} = \frac{\vec{l_t}\vec{n_t}}{\vec{l_s}\vec{n_s}} = \frac{(\vec{l_{light}} - \vec{p_t})\vec{n_t}}{(\vec{l_{light}} - \vec{p_s})\vec{n_s}} \quad (4)$$

where $m$ represents the luminosity value and is denoted as $m = el\rho n$ according to photometric stereo methods [4, 7, 41]. $e$ and $\rho$ are constants and represent light source intensity and the surface diffuse reflection coefficient, respectively. $\vec{l}$ represents the light direction and is denoted as $\vec{l} = \vec{l_{light}} - \vec{p}$, where $\vec{l_{light}}$ is the vector from the camera position to the light source position and $\vec{p}$ is the orientation vector from the camera to the spatial point. $\vec{n}$ stands for the surface normal vector and is calculated through principal component analysis (PCA) [28] using the predicted depth map. All the vectors that appear are in the camera coordinate system.

By implementing LCC, we can recalibrate the luminosity values of adjacent frames and align the colon frames, originally captured under different illumination conditions, to a uniform lighting standard. This crucial step guarantees consistent illumination between frames in colonoscopic videos. Moreover, leveraging the LCC mechanism, we reconstruct the photometric loss function $L_p(I_t, LCC \cdot \hat{I}_{s \to t})$ in ColVO to further enhance depth and pose estimation accuracy.

### 3.4 Learning Objectives for ColVO

According to [18, 26, 38], neural networks trained solely on self-supervision using colonoscopic videos can be susceptible to challenges posed by illumination variations and textureless features in the colon. To address this, ColVO incorporates both supervised and self-supervised learning into estimating depth and pose from colonoscopic videos. The supervised aspect $\mathcal{L}_{su}$ of ColVO involves minimizing the Euclidean distance between the predicted depth and pose and their GTs or references, which can be obtained using advanced sensors such as CT models [33, 44] and OptiTrack Prime [30, 31]. The self-supervised aspect $\mathcal{L}_{self}$ of ColVO involves enforcing photometric consistency and smoothness constraints on the reconstructed images and depth maps. The total loss $\mathcal{L}$ is a weighted sum of the supervised and self-supervised losses, as shown below:

$$\mathcal{L}_{su} = \mathcal{L}_{gd}(\hat{D}_t, D_t) + \mathcal{L}_{gx}(\hat{T}_{s \to t}, T_{s \to t}) \quad (5)$$

$$\mathcal{L}_{self} = \lambda_1 \mathcal{L}_p(I_t, I_s) + \lambda_2 \mathcal{L}_s(\hat{D}_t) + \lambda_3 \mathcal{L}_n(f_x) \quad (6)$$

$$\mathcal{L} = \mathcal{L}_{su} + \mathcal{L}_{self} \quad (7)$$

where $\mathcal{L}_{gd}$ and $\mathcal{L}_{gx}$ are the L1 loss in depth and pose, respectively. $\hat{D}_t$ and $\hat{T}_{s \to t}$ are the estimated depth and pose, while $D_t$ and $T_{s \to t}$ are the GTs. $\mathcal{L}_n(f_x)$ is the regularization term of PoseNet [11], which is used to prevent the network from falling into a local optimum. $\lambda$ is the parameter for each loss.

Photometric consistency is usually satisfied under the assumption that the camera is moving and the scene is stationary. However, in the colon, the colon wall can perform irregular movements and the watery colon surface can reflect the light and violate the Lambertian reflectance assumption. To address these challenges, ColVO uses two masking techniques: auto mask $\mathcal{M}_p$ [6] and temporal mask $\mathcal{M}_t$ [6]. The auto mask removes the pixels that do not conform to the photometric consistency assumption, while the temporal mask removes the pixels that do not have corresponding matches between the preceding and the following frames. The final photometric loss is as follows:

$$\mathcal{L}_p(I_t, I_s) = \mathcal{L}_p(I_t, LCC \cdot \hat{I}_{s \to t}) \odot \mathcal{M}_p \odot \mathcal{M}_t \quad (8)$$

where $LCC$ is the local color correction mechanism proposed by ColVO to handle the illumination variations in the colon.

## 4 EXPERIMENTS

### 4.1 Experimental Setup

**Datasets:**

We used two typical benchmark datasets, namely the VR-Caps dataset (VCD) and the colonoscopy simulator dataset (CSD), to train and evaluate our ColVO model. The VR-Caps dataset was collected from a virtual capsule endoscopy system [9] that provides realistic 3D models of the colon with real in-body texture mapping. We split the VCD dataset, which consists of 17,057 RGB-D images with

poses, into a training set of 8,676 continuous colon image sequences and a validation set of 8,381 colon image sequences, following [34, 39]. The CSD dataset was collected from a colonoscopy simulator [44] that offers a more challenging colon environment with rich vascular texture features.

The CSD dataset contains two paths with a total of 3500 images. We use one path with 1748 images as the training set and the other path with 1752 images as the validation set. In addition, we also reported the results on a real colon dataset [22] named olympus dataset (OD) to evaluate the generalization ability of our model.

**Evaluation Metrics:** We adopted the classic VO criterion [47] to evaluate the depth and pose. We used the error metrics (RMSE, abs.REL) and the accuracy metrics ($\delta_1$, $\delta_2$) to evaluate the depth estimation performance. We used the Absolute Trajectory Error (ATE) and Rotation Error (RE) to evaluate the error of the estimated position and rotation.

**Implementation and Settings:** We implemented our models using PyTorch [23]. Both the DepthNet and PoseNet received input of size 320×320 pixels. We trained the model for 20 epochs with a batch size of 12. We set the learning rate to $10^{-4}$ and reduced it to $10^{-5}$ after 15 epochs. Following [6, 11], we set the hyperparameters in Eq. 8 as follows: $\lambda_1 = 0.1$, $\lambda_2 = 0.001$, $\lambda_3 = 0.0002$.

## 4.2 Comparison with Other Methods

We evaluated our ColVO method against traditional VOs [5, 20] and existing end-to-end DL-based VO methods [1, 2, 17, 22, 29, 47]. The traditional multiview geometry-based VO methods include ORB-SLAM [20] and DSO [5]. As shown in Fig. 3, ORBSLAM failed to accurately predict colonoscope trajectories due to its reliance on strict feature detection and matching, which is impractical in feature-scarce colon environments. Similarly, the optical flow-based DSO algorithm also performs poorly in pose estimation. We trained these VO methods using open-source code on the VCD and the CSD dataset. However, we observed that these unsupervised VO methods failed to learn features effectively from challenging colon videos, resulting in extremely inaccurate depth and pose estimation. Therefore, we specialized them by incorporating supervised information (Eq.

5) to adapt to the challenges of the colon environment for fair comparisons. Fig. 3 and Fig. 4 show the qualitative results of the depth and pose estimation comparison with the SoTA methods, and Table 1 contains the quantitative results. Experimental results revealed that integrating supervised signals with MonoDepth2 [6] yielded competitive results in terms of depth and pose estimation. Therefore, we selected this method as the baseline method for comparison. Quantitative analysis (Table 1) revealed that the ColVO model exhibited exceptional overall performance, especially in position estimation. This advantage translates into achieving more precise colonoscope trajectories and improved 3D perception.

Visualizing the depth estimation results (Fig. 4), the ColVO method produced more accurate and realistic depth maps than the existing methods. Classical VO systems, such as SfMLearner [47] and SC-Depth [2], performed poorly in estimating the correct distances in the colon due to their focus on human activity scenes. Although the latest VO methods, such as DualRefine [1] and SRD-Depth[17], exhibited considerable capability in adapting to the colon environment, they still produced relatively coarse depth maps with noise and artifacts when faced with the more challenging CSD dataset. As for endoscopic VO systems, Endo-SfMLearner [22] and AF-SfMLearner[29] showed better overall performance in depth map prediction due to their specialized modules for handling lighting variations and low-texture images. Nonetheless, these systems suffered from suboptimal error performance partly due to their error maps demonstrating significant absolute errors. However, regarding depth estimation, ColVO's performance is slightly inferior to Endo-SfMLearner. This discrepancy arises because ColVO emphasizes improving the precision of the global camera trajectory to meet the clinical requirement of precise polyp localization. Consequently, there is a trade-off in sacrificing some local depth estimation performance. The qualitative depth prediction results are consistent with the quantitative results shown in the error metrics RMSE and abs.REL, and accuracy metrics ($\delta_1$, $\delta_2$) in Table 1.

According to the visualization of the colonoscope trajectory in Fig. 3, the trajectory predictions of ColVO are close to the ground truth and exhibit less drift or jitter compared to other methods. Most existing VO methods primarily focus on depth estimation, often

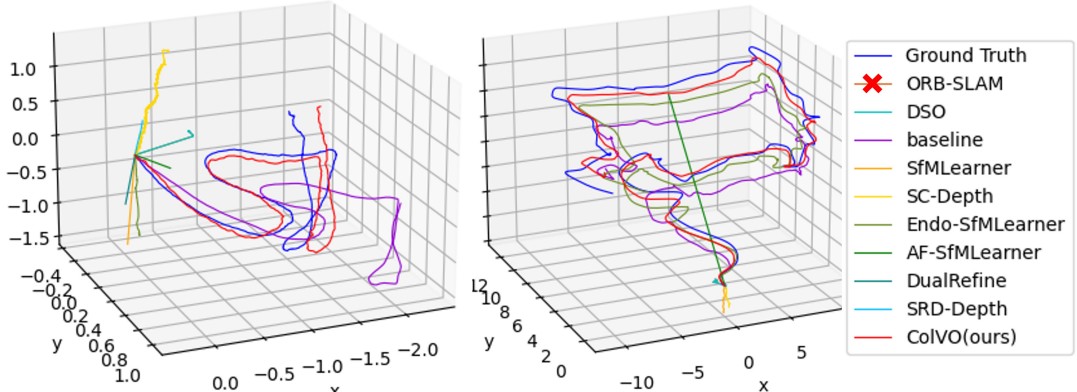

**Figure 3: Qualitative results of predicted trajectory compared with the SoTA method. Blue represents the ground truth and black represents our method.**

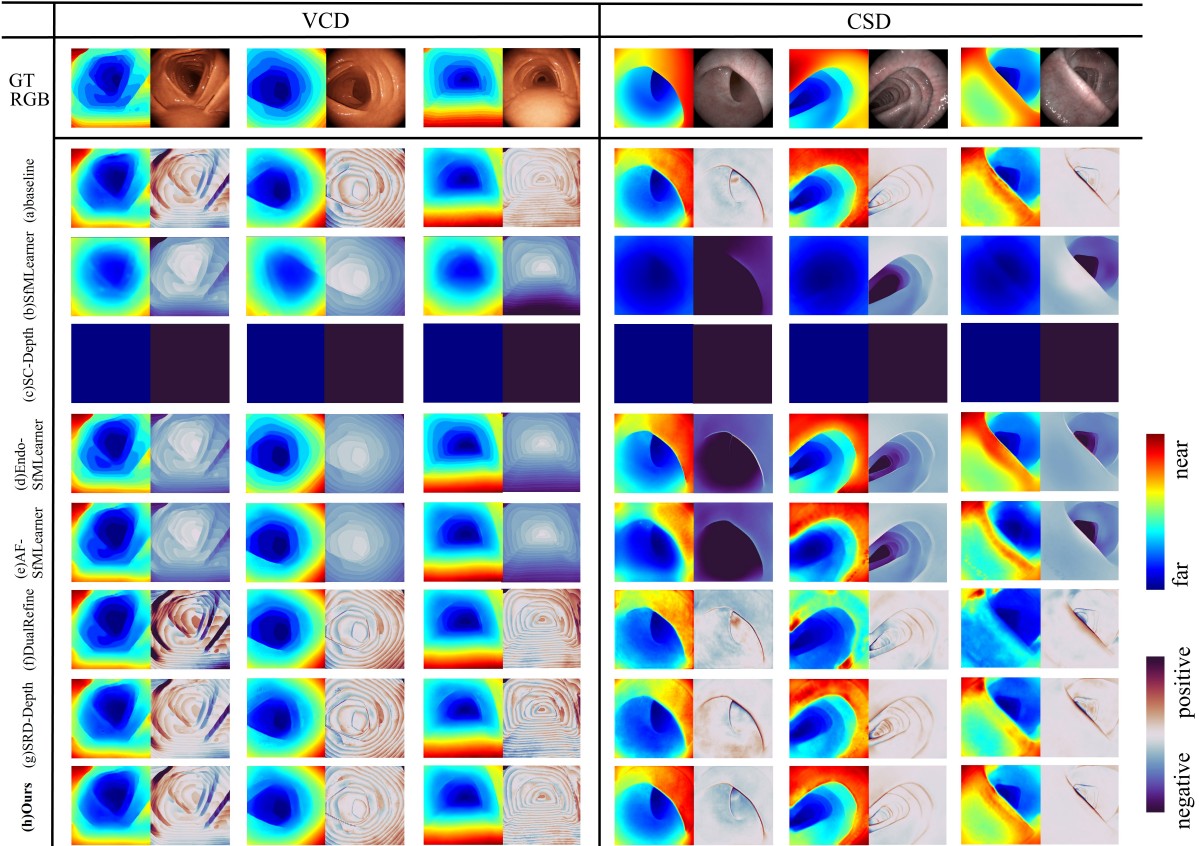

**Figure 4: Qualitative results of depth estimation compared with the SoTA methods on the VCD and CSD datasets. Each test example includes a colon image, the corresponding GT depth map, the predicted depth map, and an error map showing the discrepancy between the predicted results and GT. The depth heatmap visualizes relative depth values. The error map shows the disparities between the predicted depth with absolute scale and the GT.**

utilizing PoseNet only to assist in depth prediction. As a result, these methods are unable to accurately estimate colonoscope trajectories that align with the reference. In clinical applications, doctors typically focus more on the location of polyps, hence the importance of endoscopic pose estimation exceeds that of depth estimation. As evident from Table 2, although our method's depth estimation is slightly less accurate than the best one, its pose estimation is the best, aligning well with practical clinical applications.

## 4.3 Ablation Study

To better understand how the components of our ColVO model contribute to the overall performance in depth and pose estimation, we performed an ablation study by removing the corresponding modules from the complete ColVO model. Previous experiments demonstrated that the VCD dataset posed greater challenges to the VO task, especially in terms of colonoscope trajectory estimation. Therefore, we conducted ablation experiments on the VCD dataset. As listed in Table 2, our ColVO method outperformed the baseline method in both depth and pose estimation, demonstrating positive

and effective contributions of the designed modules in ColVO. Table 2 lists the performance degradation when specific modules are removed from our method, resulting in a decrease in performance in either depth or pose estimation, or both simultaneously. The visualization of trajectories in Fig. 5 further highlights this effect, providing a clearer depiction of the results. The purple trajectory representing the baseline exhibits the most severe deviation compared to the red trajectory of ColVO.

We analyze the benefits of each module as follows:

- **Benefits of DCDP.** Experiments in Table 2 and Fig. 5 demonstrated that this strategy led to a remarkable decline in pose error (ATE: 2.1% ↓, RE: 89.5% ↓) and a significant enhancement in scale consistency. In terms of depth estimation, the RMSE shows a slight increase. This is because the DCDP module prioritizes the accuracy of the overall camera trajectory rather than solely focusing on improving the accuracy of local depth estimation. Therefore, the DCDP can fulfill the specific clinical demand for exact polyps localization.
- **Benefits of LCC.** From the baseline to $-DCDP$ row (only LCC) in Table 2, the RMSE decreased by 5.79%, and the abs.REL by 5.95%, signifying a substantial depth accuracy enhancement due

**Table 1: Comparison with SoTA on VCD and CSD. Smaller error and greater accuracy are better. The highlight results are presented as follows: best (bold), second-best (underlined), and third-best (italicized).**

|  | Methods | Error (Pose) | | Error (Depth) | | Accuracy (Depth) | |
|---|---|---|---|---|---|---|---|
|  |  | ATE↓ | RE↓ | RMSE↓ | abs.REL↓ | $\delta_1$ ↑ | $\delta_2$ ↑ |
| VCD [9] | baseline | *0.553±0.240* | 1.605±0.687 | 0.281 | 0.058 | 98.60% | 99.82% |
|  | SfMLearner [47] | 0.476±0.191 | 1.348±0.898 | 0.887 | 0.195 | 69.86% | 91.92% |
|  | SC-Depth [2] | 0.561±0.205 | 1.851±0.813 | 1.433 | 0.400 | 41.42% | 69.75% |
|  | Endo-SfMLearner [22] | 1.956±0.628 | 0.870±0.276 | **0.242** | **0.040** | 98.81% | 99.90% |
|  | AF-SfMLearner [29] | 0.612±0.245 | 1.802±0.775 | 0.271 | *0.056* | 98.60% | 99.82% |
|  | DualRefine [1] | 0.596±0.275 | 1.709±0.807 | 0.260 | 0.054 | 98.75% | 99.83% |
|  | SRD-Depth [17] | 0.595±0.274 | *1.332±0.894* | *0.262* | 0.054 | **98.91%** | **99.92%** |
|  | ColVO(ours) | **0.475±0.240** | **0.305±0.107** | 0.264 | 0.054 | *98.77%* | 99.86% |
| CSD [44] | baseline | *5.539±3.015* | 1.862±0.622 | *0.375* | *0.078* | *96.12%* | *99.09%* |
|  | SfMLearner [47] | 6.978±3.425 | **1.374±0.524** | 1.642 | 0.282 | 55.82% | 81.58% |
|  | SC-Depth [2] | 8.491±3.293 | 1.842±0.864 | 2.297 | 0.375 | 44.51% | 67.37% |
|  | Endo-SfMLearner [22] | *5.858±2.921* | 2.116±0.766 | **0.330** | **0.059** | **97.09%** | **99.51%** |
|  | AF-SfMLearner [29] | 9.849±3.900 | 1.966±0.753 | 0.820 | 0.117 | 89.38% | 97.62% |
|  | DualRefine [1] | 8.980±3.207 | 1.464±0.870 | 0.514 | 0.106 | 88.71% | 96.68% |
|  | SRD-Depth [17] | 9.612±3.388 | *1.477±0.894* | 0.439 | 0.073 | 95.28% | 98.88% |
|  | ColVO(ours) | **3.987±1.502** | 1.735±0.762 | 0.366 | 0.077 | 96.13% | 99.10% |

**Table 2: Ablation study on components of ColVO.**

|  |  | Error (Pose) | | Error (Depth) | | Accuracy (Depth) | |
|---|---|---|---|---|---|---|---|
|  |  | ATE↓ | RE↓ | RMSE↓ | abs.REL↓ | $\delta_1$ ↑ | $\delta_2$ ↑ |
|  | baseline | 0.553±0.240 | 1.605±0.687 | 0.2813 | 0.0588 | 98.60% | 99.82% |
|  | **ColVO(ours)** | 0.475±0.240 | 0.305±0.107 | 0.2641 | 0.0546 | 98.77% | 99.86% |
|  | $-DCDP$ | 0.493±0.285 | 1.221±0.618 | 0.2650 | 0.0553 | 98.83% | 99.88% |
| Losses | $-LCC$ | 0.541±0.257 | 0.167±0.124 | 0.2821 | 0.0573 | 98.64% | 99.84% |
|  | $-\mathcal{L}_{gd}, \mathcal{L}_{gx}$ | 0.783±0.319 | 1.869±0.753 | 1.4342 | 0.4037 | 41.35% | 69.69% |
| Masks | $-\mathcal{M}_p$ | 0.628±0.329 | 1.370±0.903 | 0.2746 | 0.0563 | 98.78% | 99.86% |
|  | $-\mathcal{M}_t$ | 0.584±0.209 | 1.139±0.874 | 0.3186 | 0.0635 | 98.04% | 99.74% |

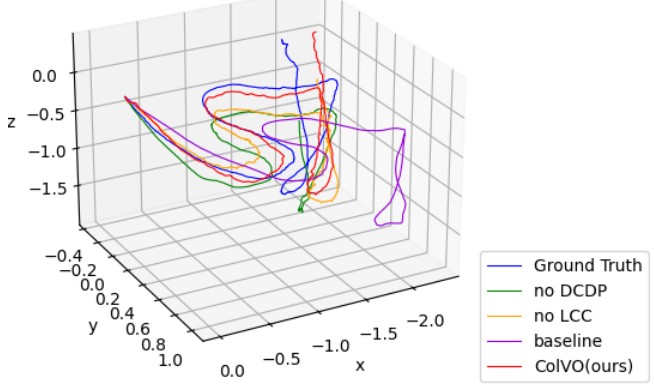

**Figure 5: Trajectory visualization of ablation experiments. The closer to the blue line, the closer to the ground truth.**

to LCC. Furthermore, pose-related errors witnessed a significant decline, with the ATE declining by 10.8%, and the RE by 23.9% compared to the baseline. Additionally, LCC is a versatile solution for environments with active light sources, like pipeline and tunnel robot operations.

- **Benefits of supervised signal.** Supervision helps ColVO to directly overcome lighting variations and texture scarcity in the colon. As listed in Table 2, without direct supervision, both depth and pose estimation results drastically decline.

- **Benefits of mask.** The auto mask $\mathcal{M}_p$ and temporal mask $\mathcal{M}_t$ can eliminate the negative effects caused by the frame-to-frame image changes and the colon's self-motion, greatly improving the ColVO's performance.

## 4.4 Generalization Ability

We validated the generalization performance of ColVO in real Olympus­Cam colon dataset using model trained on VR-Caps dataset. CT visualization in the bottom left corner of Fig. 6 revealed that the real colon exhibited a straight and elongated structure. The predicted depth map accurately captured the detailed depth map of the colon,

successfully estimating even small abnormalities within the colon, particularly at node ③. Additionally, the predicted trajectory closely aligned with the actual trends and variations of the colon, especially at node ②.

Similarly, we make a comparison between ColVO and other methods. Consequently, these methods fail to reconstruct satisfactory 3D colon models. In contrast, ColVO generates clearer and more accurate depth maps, even in polyp regions. For example, SfMLearner, SC-Depth and Endo-SfMLearner fail to produce accurate depth predictions. Although AF-SfMLearner, DualRefine and SRD-Depth show better depth maps, their trajectory predictions are unsatisfactory. The 3D colon model with complete structure and correct shape demonstrates ColVO's superior generalization performance.

## 4.5 Scale Continuity and Geometry Consistency

According to [35], scale is defined as the median of a depth map. The significant scale fluctuation between adjacent frames will cause

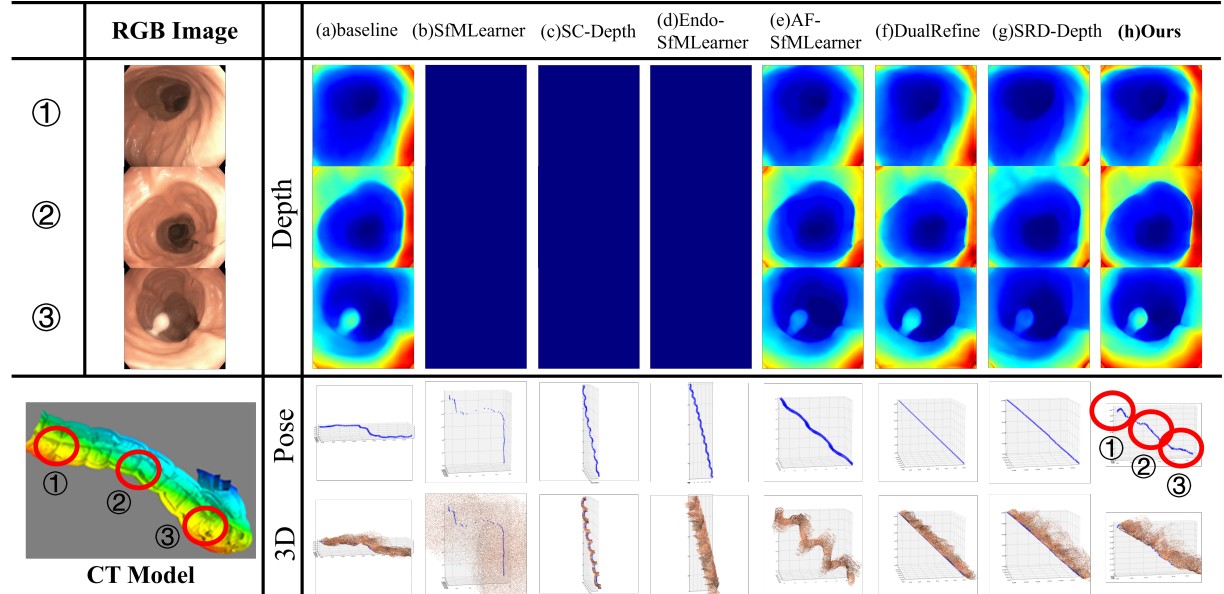

**Figure 6: Generalization validation of ColVO model on the real OlympusCam colon dataset.**

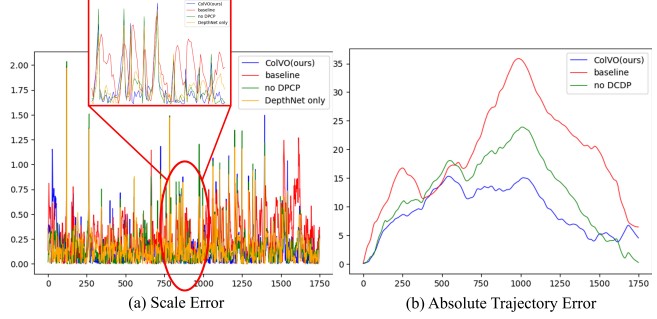

**Figure 7: Quantitative comparison in terms of depth scale results and ATE indicators with/without DCDP module.**

the corresponding depth maps fail to be accurately stitched together to produce a complete and accurate 3D colon model. Fig. 7 indicted that merging DepthNet and PoseNet in a simplistic and independent manner can actually result in depth scale estimation outcomes that are even worse than those achieved by DepthNet alone. In contrast, DCDP effectively reduced scale fluctuations and trajectory errors between frames, leading to more accurate ego-motion estimation and enhanced geometric consistency.

## 4.6 3D Visualization

To demonstrate the effectiveness of our ColVO, we implemented a 3D colon reconstruction visualization in Fig. 1 by stitching together the dense depth maps of each frame using the colonoscopic trajectory. We did not incorporate any additional design such as surface mesh, the dense point cloud generated by ColVO provided intuitive and original results. In comparison to other colon VOs [13, 18, 45], which focus only on partial colon reconstruction over

short timescales, the ColVO is capable of reconstructing the complete 3D colon model along with long timescales colonoscopic trajectory. Although there may be a few noise points present, our colon models have a clear shape and intact structure.

## 4.7 Polyp Localization

We demonstrated a valuable and practical application of our ColVO in polyp localization, as depicted in Fig. 1. Our ColVO enables immediate and direct acquisition of the 3D position of lesions during colonoscopy. This capability holds significant importance in the diagnosis and treatment of gastrointestinal diseases. After the polyps were detected, we utilized the estimated depth values and endoscope pose by ColVO to calculate the spatial positions of the polyps. By comparing with the GT values, most of the predictions can achieve a level of accuracy within millimeters.

## 5 CONCLUSION AND FUTURE WORK

This paper proposed ColVO, a novel model for simultaneous dense depth estimation and camera pose prediction in challenging colon environments. Our model leveraged two key innovations: the DCDP module, which fuses depth information with RGB and enforces geometric consistency to enhance the PoseNet performance, and the LCC mechanism, which adapts itself to the dynamic light source and improves the photometric consistency. We conducted extensive experiments on synthetic and real colon datasets and demonstrated that our model achieved superior results and clinical benefits compared to existing methods. In the future, we will focus on semantic cues such as surgical instruments, bodily fluids, polyps, and cancerous lesions can potentially serve as valuable feature cues to enhance ColVO performance.

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
