# OpenReview forum: "ColVO: Colonoscopic Visual Odometry Considering Geometric and Photometric Consistency"
_acmmm.org/ACMMM/2024/Conference — MM2024 Poster_

### Official Review · Reviewer_94r7 · 2024-05-22

**Rating:** 5
**Confidence:** 3

**Summary:**

The paper present ColVO, a method that estimates depth and camera position simultaneously. The ColVO focuses on geometric and photometric consistency to enhance accuracy of both depth and pose estimation. The ColVO include two key components：a deep couple strategy for depth and pose estimation (DCDP) and a light consistent calibration mechanism (LCC). The ColVO achieves SOTA accuracy in depth and pose estimation on synthetic dataset and real dataset.

**Strengths:**

- The proposed ColVO framework enhance depth and pose estimation accuray by introducing geometric and photometric consistency.
- SOTA is achieved on synthetic dataset and real dataset.
- The paper is generally well written.
- The problem raised by the author that the light source in the colon environment will cause brightness fluctuations between consecutive frame images as the colonoscope moves, shows that the author has a full understanding and knowledge of the application scenario he wants to study.

**Limitations:**

- Can the author make the real OlympusCam colon dataset open source?

- The author set the hyperparameters as follows:$\lambda_1=0.1, \lambda_2=0.001, \lambda_3=0.0002$. These hyperparameters correspond to forcing photometric consistency and regularization term of PoseNet, which prevent the network from falling into a local optimum. However, in the ablation experiment, the impact of regularization term of PoseNet on the entire network was not analyzed. In addition, can the author give an approximate description of how the ranges of the above three hyperparameters are determined? Are the hyperparameters used in simulated data and real data consistent? Are these hyperparameters affected by the pixel numbers of the input image? How much impact will these hyperparameter values have on the final result? Can the author add a comparative analysis under different hyperparameter values?

**Suitability:**

2

---

### Official Review · Reviewer_kYun · 2024-05-24

**Rating:** 2
**Confidence:** 3

**Summary:**

The paper poses two questions: 1) Existing methods treat colon depth and colonoscope pose estimation as independent tasks and design them as parallel sub-task branches.  2) the light source in the colon environment moves with the colonoscope, leading to brightness fluctuations among continuous frame images. Then they propose the framework ColVO which continuously estimates colon depth and colonoscopic pose using two key components, DCDP and LCC. Experiments with SoTA are shown to support the method.

**Strengths:**

1. clear description of the method, including the method to simultaneously train the two separate tasks.
2. Better visualizations of the proposed method is shown, which supports the idea.

**Limitations:**

1. About the posed Q1, the authors want.to bridge the gap with simultaneous training between two tasks, while the connection between the two tasks is not well modeled. Simply combining or concatenating them together in this task seems not to be a novel idea.
2. The proposed model seems to work better on pose estimation, while the performance on depth estimation is not the best as shown in Table. Although the authors claim that the two tasks can benefit each other, why the model does not work the best on depth estimation? This needs further explanation.
3. About the ablation study, if there are no errors in the table, the model also works well when deleting DCDP and LCC. Specifically, the "-DCDP" model works better on depth estimation, while failing to estimate accurate poses; and the "-LCC" model works well better on pose estimation, while failing to accurate depth. The authors do not provide enough about the results. Overall, the supervision signal seems the most important.
3. Various typos and problems should be corrected.

**Suitability:**

2

---

### Official Review · Reviewer_s9p3 · 2024-06-05

**Rating:** 4
**Confidence:** 2

**Summary:**

This paper aims to address the challenge of lesion localization, which involves jointly estimating depth and pose. Current methods treat pose and depth as separate tasks, but this paper suggests coupling them for better accuracy. Additionally, to tackle brightness fluctuations, the paper introduces a light-consistent calibration mechanism to filter out unreliable pixels. Experimental results confirm the effectiveness of these proposed methods.

**Strengths:**

1. The paper proposes two novel and effective models for vendor odometry.
2. The evaluation is thorough, and the baselines appear adequate.

**Limitations:**

1. I have observed that coupling pose and depth estimation leads to better results, but I still have concerns about the motivation behind coupling depth and pose estimation. While it is logical that better depth prediction improves pose prediction and vice versa, there is a concern that worse depth leads to worse poses, and vice versa. It would be helpful if the paper could provide more reasons and analysis to support the idea that depth and pose should be learned jointly.

2. Regarding equation (4) for the LCC term, I find it difficult to interpret. As I understand, p represents the viewing vector and l_light represents the light direction. However, I am unable to clearly interpret the geometry of the light transport. It would be beneficial to include a figure for better understanding.

3. In the minor issues, I suggest replacing "Fov" with "FoV" in line 92.

**Suitability:**

2

---

### Meta-Review · Area_Chair_2kjU · 2024-07-01

**Recommendation:** Accept (Poster)
**Confidence:** 5

**Metareview:**

This paper initially got mixed reviews. The authors' rebuttal addressed the reviewers' concerns and all reviewers reached a consensus on the acceptance of the paper.